# DeepExposure: Learning to Expose Photos with Asynchronously Reinforced Adversarial Learning

**Runsheng Yu**[*]
Xiaomi AI Lab
South China Normal University
runshengyu@gmail.com

**Wenyu Liu** [*]
Xiaomi AI Lab
Peking University
liuwenyu@pku.edu.cn

**Yasen Zhang**
Xiaomi AI Lab
zhangyasen@xiaomi.com

**Zhi Qu**
Xiaomi AI Lab
quzhi@xiaomi.com

**Deli Zhao**
Xiaomi AI Lab
zhaodeli@xiaomi.com

**Bo Zhang**
Xiaomi AI Lab
zhangbo@xiaomi.com

## Abstract

The accurate exposure is the key of capturing high-quality photos in computational photography, especially for mobile phones that are limited by sizes of camera modules. Inspired by luminosity masks usually applied by professional photographers, in this paper, we develop a novel algorithm for learning local exposures with deep reinforcement adversarial learning. To be specific, we segment an image into sub-images that can reflect variations of dynamic range exposures according to raw low-level features. Based on these sub-images, a local exposure for each sub-image is automatically learned by virtue of policy network sequentially while the reward of learning is globally designed for striking a balance of overall exposures. The aesthetic evaluation function is approximated by discriminator in generative adversarial networks. The reinforcement learning and the adversarial learning are trained collaboratively by asynchronous deterministic policy gradient and generative loss approximation. To further simply the algorithmic architecture, we also prove the feasibility of leveraging the discriminator as the value function. Further more, we employ each local exposure to retouch the raw input image respectively, thus delivering multiple retouched images under different exposures which are fused with exposure blending. The extensive experiments verify that our algorithms are superior to state-of-the-art methods in terms of quantitative accuracy and visual illustration.

## 1 Introduction

Retouching raw low-quality photos into high-quality ones will greatly increase the aesthetic experience of our vision. Due to the requirement of expertise of photography, photo quality enhancement is beyond the capability of non-professional users, thus leading to the new trend of automatic techniques of image retouching.

The traditional methods for automatic image retouching include retinex of a theory based on human image perception [21], transform of using enhancement-parametric operators to retouch images [1], and exposure/contrast fusion [10, 23, 25]. But these methods have their own limitations: Most of them are incapable of comprehending semantic information or object relationship in images well. With the prevalence of deep learning, many researchers focus on applying this method to the image retouching area. In general, the image retouching approaches based on deep learning fall into three categories:

---

[*]Joint first authors.

1) *The transfer methods:* Many researchers regard image retouching as style transfer, including domain transfer and image-to-image translation [17, 18, 16, 15, 22, 5]. In light of the principle of generative adversarial network [12], these methods provide a novel perspective on image quality enhancement. However, one of challenges is to derive photo-realistic effects for these generation-based approaches. 2) *Retouching-driven methods:* These methods focus on generating retouched photos directly from input low-quality images [11, 6, 4, 39, 36]. Various losses are deliberately designed to enhance image from different perspectives, e.g. SSIM loss, texture loss and color loss. 3) *The sequence-based methods:* The sequence-based methods are to generate an operation sequence which can be clearly understood by human [14, 38, 29, 7, 37, 40]. One kind of these approaches is to employ reinforcement learning [14, 38, 29]. Hu *et al.* [14] and Park *et al.* [29] utilized deep reinforcement learning (DRL) to generate human-understandable operation sequences while Yang *et al.* [38] applied DRL to generate personalized real-time exposure control.

When retouching a photo, we frequently encounter difficulty that some parts of a photo are too dark while other parts are too bright due to the limitation of photographic techniques or hardware, especially for small camera modules embedded in mobile phones. This exposure issue cannot be easily addressed by the global adjustment since the required adjustment operations vary in different areas. Professional photographers always employ the exposure-blending with luminosity masks to perform image post-processing [20]. That is, they create different layer masks for different objects and adjust each of objects independently. By virtue of this skill, they can cope with those scenarios with a wide range of illumination distribution flexibly. Nevertheless, it may not be easily used in automatic retouching: Since the different objects in an image are semantically correlated, we need to consider the inherent relationship on illuminations and colors for each local exposure operation discreetly when designing algorithms. In addition, it is of great significance to find an appropriate metric to evaluate whether a photo is aesthetically good or not. However, the traditional image evaluation metrics may not work well in this situation [32].

In this paper, we develop a reinforced adversarial learning framework to solve these problems. The deep reinforcement learning is exploited to learn multiple local exposure operations and an adversarial learning method is harnessed to approximate the Aesthetic Evaluation (AE) function, i.e. an evaluation method to judge the subjective quality of an image. Both reinforcement learning and adversarial learning can be trained together as a whole pipeline by asynchronous deterministic policy gradient and generative loss approximation.

Our main contributions are summarized as follows:

1. Based on deep reinforcement learning, we propose an exposure-blending-based framework which can retouch local areas of images flexibly with only exposure operation.

2. There exists a non-differential operation in the whole process. We leverage the generative loss approximation to make it differential and asynchronous learning to make the learning process stable.

3. By asynchronously reinforced adversarial learning, we propose an approach to training both exposure operations and aesthetic evaluation function. The asynchronous update policy gradients aid the algorithm in tuning sequential exposures while adversarial learning facilitates to learn aesthetic evaluation function.

4. The whole pipeline proceeds with image-unpaired training and can be efficiently performed with super-resolution in practice. Our algorithm do not directly generate any pixels and can well preserve the details of the original image.

5. For computers of limited memory when training, we devise an algorithm to reuse the discriminator as the value function, effectively reducing memory occupation and accelerating the training speed.

## 2   Methodology

In this section, we will present the details of our algorithms in five sub-sections: the problem formulation, the asynchronous deterministic policy gradient, the adversarial learning for AE function, the generative loss approximation and the alternative form of value function.

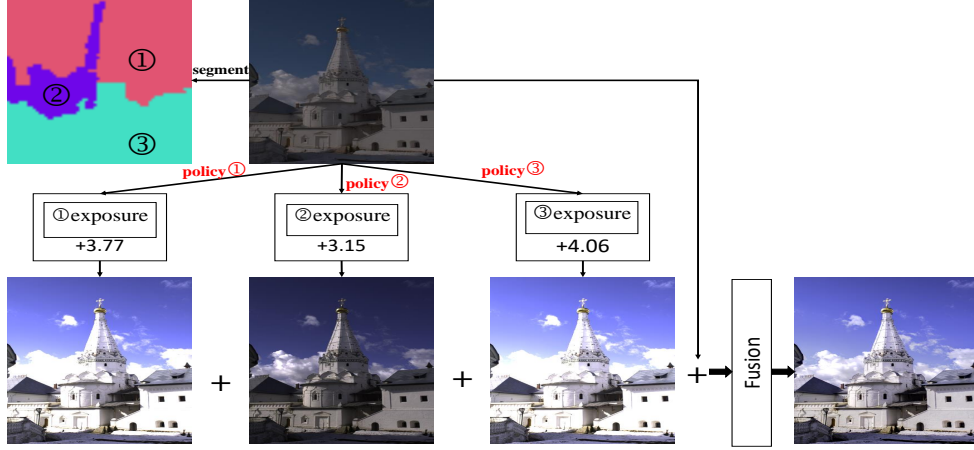

Figure 1: The schematic illustration of our algorithm. Firstly, we harness image segmentation to obtain sub-images. For different sub-images, we use different exposures according to the policy network and they are fused together to form the final high-quality image.

## 2.1 Problem formulation

Let $a_t$ denote the $t$-th exposure operation. For reinforcement learning, $a_t$ is the action at step $t$. Thus, $\mathcal{A} = \{a_0, a_1, \ldots, a_T\}$ forms the action space, which is the sequential exposure operations in the scenario of image retouching. The local exposure image retouching can be formulated as follows:

$$\arg \max_{\mathcal{A}} \phi(P_T(s_0, \mathcal{A})), \tag{1}$$

$$P_T(s_0, \mathcal{A}) = EB \circ a_T \circ a_{T-1} \circ a_{T-2} \cdots \circ a_0 \circ s_0, \quad S^f = P_T(s_0, \mathcal{A}) \tag{2}$$

where $\phi(\cdot)$ is the aesthetic evaluation function, "$\circ$" is the function composition operation, $EB$ is the exposure blending manipulation, and $s_0$ is the first state; $S^f$ is the final image after fusion. Our algorithm is to find the optimal operations which maximize the $\phi(\cdot)$ function. In order to solve this problem, there are two sub-problems to be considered: 1) how to derive the optimal sequential operations $\mathcal{A}$ through learning methods, and 2) how to model the AE function $\phi(\cdot)$.

Since each exposure operation needs to consider the original low-resolution input image $s_0$, the local fusion image $s_t^l$ at step $t$, and the $t$-th segment sub-image $seg_t$, we define the state $s_t$ as $s_t = \{s_t^l, seg_t, s_0\} \in \mathcal{S}$, where the sub-images stem from image segmentation $\{seg_0, seg_1, \ldots, seg_T\} = segment(s_0)$ and $\mathcal{S}$ is the state space. In order to express the probability of an interaction, we utilize the transition function to describe the exposure operation as $s_{t+1} = a_t \circ s_t = p(s_t, a_t)$. The EB denotes a function that integrates all the global filter images $[S_0^g, S_1^g, \ldots, S_n^g]$ together, $EB : [S_0^g, S_1^g, \ldots, S_n^g] \rightarrow S^f$ .

For sub-problem 1, since the operation $a_t$ is calculated according to its previous state $s_t$, we can regard sub-problem 1 as agent-environment interaction problem, i.e. the Markov Decision Process (MDP) problem. Here, we take the reinforcement learning is one of the feasible tools to deal with that problem.

According to the nature of reinforcement learning, it is plausible to determine the reward function with the AE function to evaluate how action $a_t$ performs. One thing that needs to consider is that due to the limitation in the image retouching area, it is hard to obtain the intermediate process and we only have the terminating results to obtain the rewards, also known as the sparse reward issue. Thus, the AE function applies only in the final step. Here, we define the reward function $r_t(s_t, a_t)$ as

$$r_t(s_t, a_t) = \begin{cases} 0, & t \neq T \\ \phi(P_T(s_0, \mathcal{A})), & t = T. \end{cases} \tag{3}$$

From Eq. (3) we can see that the reinforcement learning employed in our image retouching framework does not need to consider the intermediate rewards, thereby simplifying the overall reinforcement learning procedure. Therefore, we can write the summation of discounted reward $r_0^\gamma$ (or return function) as $r_0^\gamma = \sum_{t'=0}^T \gamma^{t'} r_{t'}(s_{t'}, a_{t'}) = r_T(s_T, a_T) = \phi(P_T(s_0, \mathcal{A}))$. To proceed, we use the advantage actor-critic framework as our basic reinforcement learning model [35].

Since the exposure operations are decided by the current state, we can define the policy $\pi : \mathcal{S} \to P(\mathcal{A})$ and discounted state visitation distribution $\rho^\pi$ to model this process. With these definitions, we can cast the maximization of the AE function as the optimization

$$\arg\max_\pi J(\pi) = \arg\max_\pi \mathop{E}_{s \sim \rho^\pi, P_T \sim \pi} [r_T | \pi]. \tag{4}$$

Similarly, we use the value function

$$V^\pi(s) = \mathop{E}_{s \sim \rho^\pi, P_T \sim \pi} [r_T] \tag{5}$$

to evaluate the states. Also, we harness the state-action value function $Q^\pi(s_t, a_t) = \mathop{E}_{s \sim \rho^\pi, a \sim a_t, P_T \sim \pi} [r_t(s_t, a_t) + \gamma V^\pi(p(s_t, a_t))]$ and its normalization form $A^\pi(s, a) = Q^\pi(s, a) - V^\pi(s)$ to determine the action $a_t$ at that state $s_t$ to reduce high variability. The value function can be regarded as the proxy of the AE function.

The AE function and value function can be estimated by temporal difference method [33] that can be formulated with

$$L_V = \frac{1}{2} \mathop{E}_{s \sim \rho^\pi, a \sim \pi(s)} [\delta^2], \ \delta = r_t + \gamma V^\pi(p(s_t, a_t)) - V^\pi(s_t). \tag{6}$$

Through the equation above, we can use the AE function to guide the value function $V$ by minimizing Eq. (6).

The value function $V$ and the policy function $\pi$ can be approximated by neural networks $V^\omega$ and $\pi^\theta$ respectively, where $\omega$ and $\theta$ are learnable parameters. Thus we can take advantage of learning method to approximate these two functions. Since the operation is continuous, we employ the deterministic policy gradient (DPG) theorem [31] to update our model

$$\nabla_\theta J(\pi_\theta) = \mathop{E}_{s \sim \rho^\pi} [\nabla_\theta \pi_\theta(s) \nabla_a A^\pi(s, a; \theta_t) | a = \pi(s)], \tag{7}$$

$$\theta_{t+1} = \theta_t + \beta \mathop{E}_{s \sim \rho^\pi} [\nabla_\theta \pi_\theta(s) \nabla_a A^\pi(s, a; \theta) | a = \pi(s)], \tag{8}$$

$$\text{and } \omega_{t+1} = \omega_t + \alpha \mathop{E}_{s \sim \rho^\pi} [[r(t) + \gamma V^\pi(p(s_t, a_t); \omega) - V^\pi(s_t; \omega)] \nabla_\omega V^\pi(s_t; \omega))], \tag{9}$$

where $A$ can be calculated by the normalization form equation mentioned above. Here, we can use $V$ function to get all the equations mentioned above.

## 2.2 Asynchronous deterministic policy gradient

From the common point of view, a variety of reinforcement learning algorithms including DPG need the assumption that the samples are independently and identically distributed [24]. However, the sequential data from practical tasks usually violate the assumption, which is also known as high temporal correlations. The technique of experience replay can elaborately circumvent the problem [27]. But for our case, we need to accomplish the whole process without interruption and it is memory consumption to store transitions $(s_t, a_t, r_t, s_{t+1})$ since each one state contains many images. Therefore, the commonly used experience replay or out-of-order training method [14] may not be suitable here. Under this circumstance, we choose to update our actor network and the critic network by virtue of asynchronous updating. The update formulas can be approximated as

$$\mathop{E}_{s \sim \rho^\pi} [\nabla_\theta \pi_\theta(s) \nabla_a A^\pi(s, a) | a = \pi(s)] \approx \frac{1}{TN} \sum_{i=0}^{N} \sum_{t=0}^{T} \nabla_\theta \pi_\theta(s_{it}) \nabla_{a_{it}} A^{\pi_\theta}(s_{it}, a_{it} = \pi_\theta(s_{it})), \tag{10}$$

$$\mathop{E}_{s \sim \rho^\pi} [[r + \gamma V^\pi(p(s, a); \omega) - V^\pi(s; \omega)] \nabla_\omega V^\pi(s; \omega))] \approx \frac{1}{TN} \sum_{i=0}^{N} \sum_{t=0}^{T} [r_i(t) + \gamma V^\pi(p(s_{it}, a_{it}); \omega)$$
$$- V^\pi(s_{it}; \omega)] \nabla_\omega V^\pi(s_{it}; \omega)), \tag{11}$$

where $N$ is a mini-batch size and $T$ is the sequence length. With this asynchronous update method, we can reduce the effect of high temporal correlations. Eq. (10) reveals how to calculate the gradient

of parameter $\theta$ while Eq. (11) shows how to calculate the gradient of parameter $\omega$. The retouching processes can also be done by $N$ threads in parallel, then our asynchronous update method is the continuous control of asynchronous policy gradient similar to the framework in [26].

This asynchronous update method can have the same effect as the replay buffer. More information can be found in Appendix A.

### 2.3 Adversarial learning for the AE function

In the preceding section, we propose a method to find the optimal sequential operations. Nevertheless, there still exists one problem: How to get the reward function without knowing the AE function $\phi(\cdot)$. One simple way is to learn this AE function by neural networks. Inspired by generative adversarial network [12], we treat the AE function as the discriminator and learn through adversarial learning. In this case, we use the Wasserstein GAN as our adversarial learning framework [2].

Let $p_d$ denote the distribution of the expert retouched images and $p_a$ the distribution of our algorithm-retouched images. According to [2], we define the loss of discriminator as

$$L_D = E_{\tilde{S}^f \sim p_d}[D^\beta(\tilde{S}^f)] - E_{S^f \sim p_a}[D^\beta(S^f)] + \lambda E_{\hat{S}^f \sim p_{\hat{S}^f}}[(\|\nabla_{\hat{S}^f} D^\beta(\hat{S}^f)\|_2 - 1)^2], \quad (12)$$

where $\beta$ is the parameter of the discriminator and $\hat{S}^f = \epsilon S^f + (1 - \epsilon)\tilde{S}^f$ and $\epsilon \in [0, 1]$. The gradient penalty is applied to ensure that $D^\beta$ is Lipschitz-continuous [13]. The discriminator is designed to discriminate whether the photos are retouched by an expert or by our own method: $D^\beta(\tilde{S}^f) = \phi(P_T(s_0, \mathcal{A}))$. Thus, it can be leveraged as the AE function.

### 2.4 Generative loss approximation

Normally, the algorithmic framework of adversarial learning needs to combine a generator with the discriminator end-to-end. For our problem, however, the loss function of the original generator cannot be adopted because $EB$ step in our pipeline is non-differential. For this reason, we opt to approximate the original generative loss gradient through the DPG in Eq. (7), i.e.

$$\nabla_\theta J(\pi_\theta) \approx C \nabla_\theta L_G, \quad (13)$$

where $L_G$ is the loss of the generator function and $C$ is a positive constant number. And if $C$ is conformed to the learning rate, the gradient descent between DPG and GAN losses is equivalent. This approximation helps us solve the non-differential problem. More details can be found in Appendix C.

### 2.5 Alternative form of value function

To diminish the sparse reward problem, we can leverage the discriminator $D^\beta$ to replace $V^\omega$ by solving the value function $L_V$ directly

$$V(s_t) = D(P_T(s_0, \mathcal{A})) * \gamma^{-t+T}, \quad Q(s_t, a_t) = D(P_T(s_0, \mathcal{A})) * \gamma^{-t+1+T}, \quad (14)$$

$$\text{and } A^\pi(s_t, a_t) = D(P_T(s_0, \mathcal{A})) * (\gamma - 1) * \gamma^{-t+T}. \quad (15)$$

Now one more question is pending to be answered: Since $P_T(s_0, \mathcal{A})$ contains future information difficult to be obtained, we cannot attain $P_T(s_0, \mathcal{A})$ directly. But if the sub-images are not so many, then we can use the intermediate state $S_t$ to approximate the final step, say, $S_t \approx P_T(s_0, \mathcal{A})$. This approximation is plausible in our scenario because the discriminator admits an approximately exponential decay of importance with respect to the time dimension of parameter updating. The closer the time step is to the final step, the larger weights the reward obtains. From this perspective, the formulation is conformal to intuition as well.

The surrogate of value function will reduce memory occupation and the training time since one of deep neural networks no longer needs re-training. De facto, this form is like the reward-shaping method [28] and more details can be found in Appendix B.

## 3 Exposure blending

Our algorithm will yield an exact exposure value for each segmentation from policy network. The exposure value will be applied for the entire image, thus resulting in multiple retouched images of

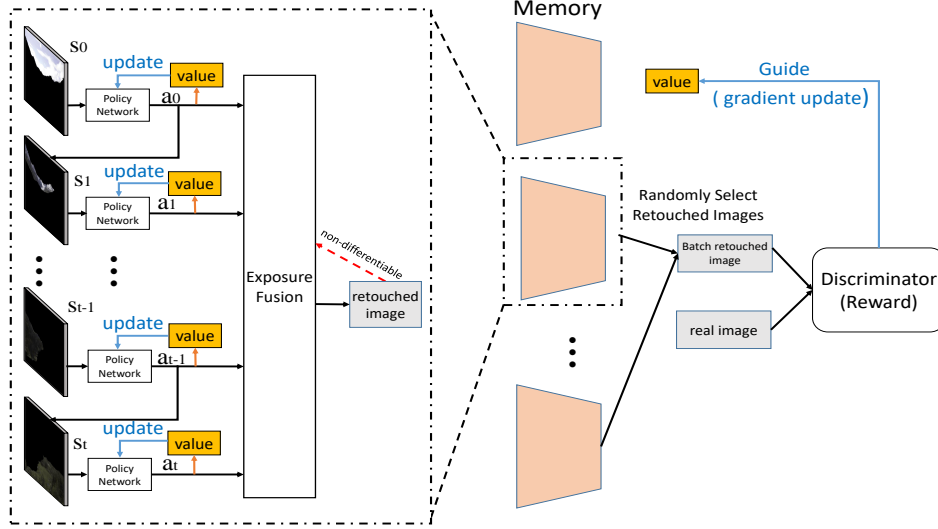

Figure 2: The pipeline of our algorithm. The first input state $s_0$ consists of two parts: The first segmentation sub-image $seg_0$ as well as the raw low-resolution image $S_0$. The policy network calculates the exposure value $e$ and the action is the whole process that generates the locally exposed image $S^l$ as well as the globally exposed image $S^g$ (details can be found in Algorithm 3 in Appendix E). The value function is used to evaluate the action. We only update the value and policy gradients when finishing a mini-batch of images retouching with the asynchronous update method. The processed images will be restored in memory. The discriminator is trained by randomly selecting a batch of the algorithm-retouched images and the expert-retouched data unpairedly, and it will guide the value function to update.

different exposures for one input image with some visual artifacts. To enhance the final visual effect, we harness the blending approach in High Dynamic Range (HDR) Imaging for exposure fusion [25]. We find the well-exposed areas and blend them together through pyramidal image decomposition

$$L(S_o^{ij})^k = \sum_{l=1}^n Gauss(w_{ij}^l)^k L(S_{ijl}^g)^k \qquad (16)$$

where $L(S_o^{ij})^k$ is the $k$-level Laplacian pyramid decomposition at pixel $i$ and $j$, $Gauss(w_{ij}^k)$ is the $k$-level Gaussian pyramid of weights at pixel $i$ and $j$, and $L(S_{ijl}^g)^k$ is the $k$-level Laplacian pyramid decomposition of the $l$-th exposed image at pixel $i$ and $j$. In fact, this method can be regarded as pseudo multiple exposure fusion. More details can be found in Appendix D.

## 4 The algorithmic pipeline

The whole pipeline of our algorithm is presented as follows:

As shown in Figure 2, we first use the image segmentation method to segment the whole image into several sub-images (all the training images are of size $64 \times 64 \times 3$ in our experiment). During the action-generating stage $t$, we concatenate the input low-resolution image $S_0$, the sub-image $seg_t$, and the direct fusion image $S_t^l$ as the state $s_t$. Then, a policy network is exploited to compute different exposures that are applied on the image locally and globally. The local filter can be formed as $S_{t+1}^l = S_t^l \odot bg\_mask + e_t * seg_t$ ,where $seg_t$ is the $t$-th sub-image, $bg\_mask$ is the background mask which does not need exposure at this step, $e_t$ is the corresponding exposure value, $\odot$ means element-wise product, and $*$ is scalar-matrix multiplication. The global filter performs $S_{t+1}^g = S_0^g * e_t$ and all the global filters operate on the original image $S_0$. When finishing $s_{t+1}$, we apply the value function to evaluate the quality of this step by calculating the one-step gradient using Eq. (8). After all the sub-images are processed, we blend all the images of different exposures and the input image together. The exposure fusion is made with Eq. (16).

To update policy network and value network through Eq. (10) and Eq. (11), we repeat trials of 8 mini-batches to collect robust gradients. For the discriminator, we randomly choose a mini-batch of machine-retouched and expert-retouched photos to train the discriminator through Eq. (12). Due

to the advantage of GAN, the training process can be unpaired for two kinds of input images. This unpaired training can avoid the difficulty of acquiring the paired data in real environment. So as to make the discriminator more reliable, we take the similar method in [14]: The contrast, saturation and illumination features are extracted and then are concatenated with the retouched RGB image together, finally forming a $(3 + 3)$-channel image as the input of the discriminator. Moreover, as we find in our experiment, the discriminator reward can be applied directly to the finally local direct fusion image $S_T^l$ to improve the effectiveness of our model.

In the test stage, as Figure 1 shows in Appendix F, a raw image $I_0$ of arbitrary size is resized to $64 \times 64$. The resized image is fed into the policy network to derive exposure values and global filters, as well as local filters. These variables are used to retouch the resized image for intermediate computations. For the test image $I$ of original size, only global filters will be employed. After all exposure values are solved from the sub-images and applied to generate the re-exposed image $I_i$, the final retouched image of original size is blended by $\{I_0, I_1, ...I_t\}$ using Eq. (16).

The pseudo-codes are presented in Appendix E.

## 5 Experiment

### 5.1 Implementation details

We train our model on MIT-Adobe FiveK [3], a dataset which contains 5,000 RAW photos and corresponding retouched ones edited by five experts for each photo. To perform fair comparison with state-of-the-art algorithms, we follow the experimental protocol presented by Hu *et al.* [14]. We separate the dataset into three subsets: $2,000$ input unretouched images, $2,000$ retouched images by expert C, and $1,000$ input RAW images for testing. Unless noted otherwise, all the images in training and testing stages are scaled down to $500$ along long edge.

The architecture of our networks is detailed in Figure 2 in Appendix F. Specifically, our model is differentiable according to the form of value function. If the value function is calculated directly, the value function is approximated by the discriminator, hereinafter referred to DeepExposure II. Otherwise, the value function is learned with neural networks that we depict in Appendix F, hereinafter referred to DeepExposure I. All the networks are optimized by Adam [19].

Here we present some details between different networks. For discriminator network, the original learning rate is $5 \times 10^{-5}$ with an exponential decay to $10^{-3}$ of the original value. The batch size for adversarial learning is 8. For policy network, the original learning rate is $1.5 \times 10^{-5}$ with an exponential decay to $10^{-3}$ of the original value. The Ornstein-Uhlenbeck process [34] is used to perform the exploration [2]. The mini-batch size for policy network is 8. The parameter will not be updated until a collection of parameters are obtained. For value network, if it is DeepExposure I, the original learning rate is $5 \times 10^{-4}$ with an exponential decay to $10^{-3}$ of the original value. Otherwise, we do not use the value network for DeepExposure II. The $\gamma$ parameter is set $0.99$.

For image segmentation, we take advantage of the graph-based method to segment images [8]. Since this segmentation is performed according to texture and color in an unsupervised manner, it will provide policy network with the low-level information.

The codes are run on P40 Tesla GPU. DeepExposure I takes about 320 min to converge while DeepExposure II takes 280 min. All the networks are implemented via Tensorflow.

### 5.2 Experimental results

The quantitative results of our models are obtained on the test dataset of MIT-Adobe FiveK. The compared baseline and state-of-the-art methods include the sequence-based method of Exposure [14], the unpaired style transfer method of CycleGAN [41], the fusion-based retouching method of FI [10], and the paired image enhancement method of DPED [16].

From Table 1, we can see that our method consistently outperforms the involved algorithms. It is worth noting that compared with the Exposure algorithm that is also established on reinforcement learning, our algorithm attains higher scores in both MSE and PSNR using only one filter. This success exhibits the power of local operations in the asynchronous mode. As an adaptive exposure fusion method, our method outperforms FI that is the state-of-the-art algorithm of single image fusion,

Table 1: Quantitative comparison of compared algorithms on MIT-Adobe 5K test dataset. For MSE (Mean Squared Error), the smaller number the better. For PSNR (Peak Signal-to-Noise Ratio), the larger number the better. The best results are highlighted in bold fonts. DE is our method, meaning the abbreviation of DeepExposure. DeepExposure I (DE I) learns both value and discriminator network. DeepExposure II (DE II) employs the alternative form of value function.

| Metric | Exposure [14] | CycleGAN [41] | DPED [16] | FI [10] | DE I | DE II |
|--------|---------------|---------------|-----------|---------|------|-------|
| MSE    | 97.99         | 101.10        | 99.04     | 105.2   | **95.44** | 96.42 |
| PSNR   | 28.27         | 28.12         | 28.20     | 27.92   | **28.38** | 28.33 |

indicating that learning-based exposure with reinforcement is better than empirically crafted exposure. One more thing is that our algorithm is an unpaired one but superior to the paired method DPED, showing the effectiveness of our method in the unpaired setting.

Figure 3 illustrates some imagery examples of our algorithms and other state-of-the-art methods. Except for the methods mentioned above, we also compare Deep Photo Enhancer [5] and Deep Guided Filter [36] which are the latest relevant work[3]. We can find that among all compared approaches, our method restores the details of original images better and enhance the saturation more effectively.

Due to the limitation of space, we cannot show more results of our experiments. We conclude some key features here: 1) As the Exposure approach does, our method can deal with higher resolution images than other methods. The detailed experiments can be found in Appendix G1. 2) Our algorithm does learn to adapt various exposure parameters to sub-images of various styles. The detailed experiments can be found in Appendix G2. We also demonstrate more retouching results in Appendix G3.

## 6 Conclusion

We develop a reinforced adversarial learning algorithm to learn the optimal exposure operations of retouching low-quality images. Comparing with other methods, our algorithm can restore most of the details and styles in original images while enhancing brightness and colors. Moreover, our method bridges deep-learning methods and traditional methods of filtering: Deep-learning methods serve to learn parameters of filters, which makes more precise filtering of traditional methods. And traditional methods reduce the training time of deep-learning methods because filtering pixels is much faster than generating pixels with neural networks.

Our algorithm relies on image segmentation to mimic the layering of illumination mask. Therefore, using semantic segmentation algorithms instead of unsupervised one may improve the capability of learning exposures. One natural extension of our algorithm is to combine with other usual filters, such as the contrast filter and the tone filter, which is left for future work. One limitation is that due to the sparse reward, the local exposure might not be exactly accurate value. Thus other novel methods like curriculum learning [9] or curiosity-driven learning [30] will be explored in the future.

## 7 Acknowledge

The authors would like to thank Haorui Zhang, Kaiyuan Huang, Suming Yu, and Zhenyu Shi for providing some guides on professional photography and reinforcement learning. We also thank Yuanming Hu for giving us lots of guides. We are grateful to the anonymous reviewers for the insightful comments.

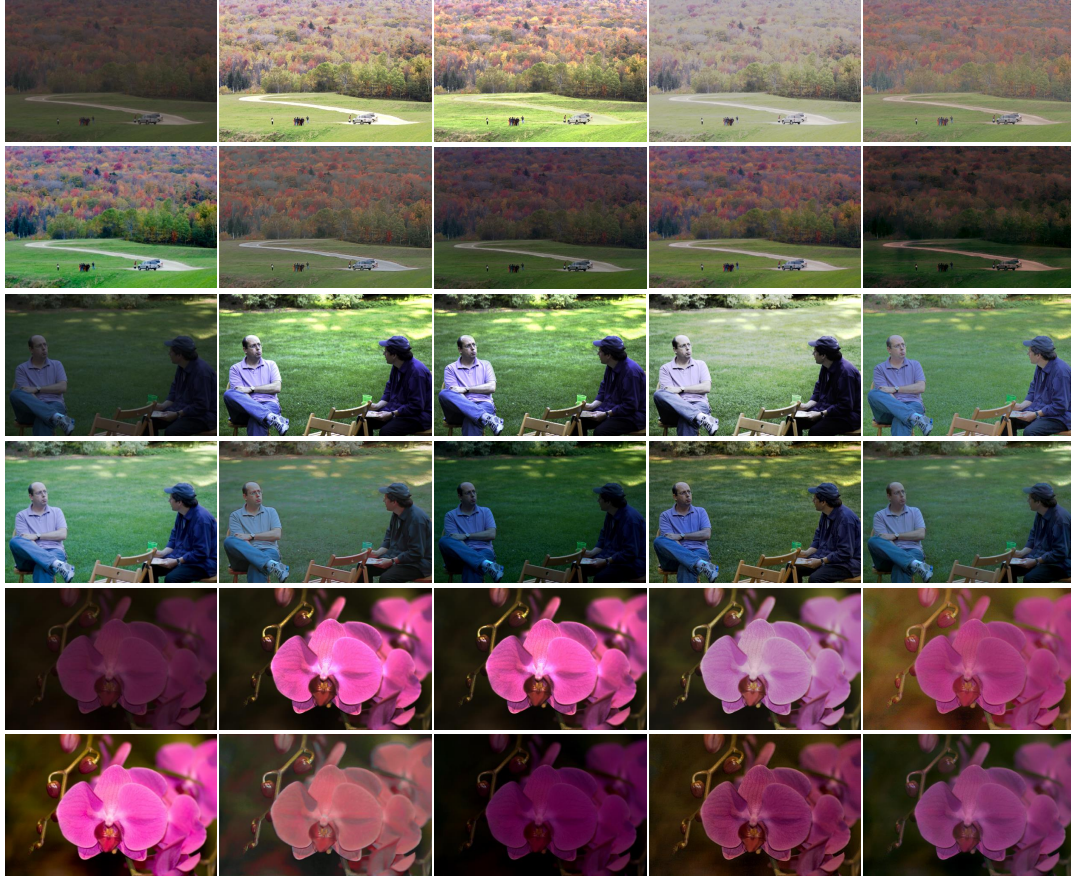

Figure 3: Retouched images of different algorithms. From left to right, top to bottom: Original input image, our DeepExposure I, DeepExposure II, Exposure [14], FI [10], Expert C, DPED [16], CycleGAN [41], Deep Photo Enhancer [5] and Deep Guided Filter [36].

## Footnotes

[2] The Ornstein-Uhlenbeck process is $dx_t = \zeta(\mu - x_t)\,dt + \sigma\,dW_t$, where $W_t$ is the Wiener process and $\zeta, \mu, \sigma$ are the hyper-parameters. We set the hyper-parameters as $\zeta = 0.015$, $\mu = 0$, and $\sigma = 0.03$.

[3] Due to the limited availability of the codes during this work, we are not capable to access all the source codes. Therefore, for different methods, we utilize different strategies: For methods Exposure, FI, and DPED, we use the already-trained model to run the results; for CycleGAN, we retrain it with MIT-Adobe 5K training dataset; for Deep Guided Filter and Deep Photo Enhancer, we use demos from their own websites (`http://wuhuikai.me/DeepGuidedFilterProject/` and `http://www.cmlab.csie.ntu.edu.tw/project/Deep-Photo-Enhancer/`, respectively) to derive imagery results.

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
