[Supplementary Material · supplementary.pdf]

# DeepExposure: Learning to Expose Photos with Asynchronously Reinforced Adversarial Learning Supplementary Materials

**Runsheng Yu**[*]
Xiaomi AI Lab
South China Normal University
runshengyu@gmail.com

**Wenyu Liu** [*]
Xiaomi AI Lab
Peking University
liuwenyu@pku.edu.cn

**Yasen Zhang**
Xiaomi AI Lab
zhangyasen@xiaomi.com

**Zhi Qu**
Xiaomi AI Lab
quzhi@xiaomi.com

**Deli Zhao**
Xiaomi AI Lab
zhaodeli@xiaomi.com

**Bo Zhang**
Xiaomi AI Lab
zhangbo@xiaomi.com

## Appendix

### A. Experience buffer and asynchronous policy gradient

In this section we will discuss how our asynchronous update method can replace the experience buffer.

The $V$-value function is as follows

$$V^\pi(s) = \underset{s\sim s_0, t\sim\pi}{E}[r_0^\gamma]. \tag{1}$$

The updates for online $V$-value and policy network function can be written as

$$\omega_{t+1} = \omega_t + \alpha[r(t) + \gamma V^\pi(p(s_t, a_t); \omega_t) - V^\pi(s_t; \omega_t)]\nabla_\omega V^\pi(s_t; \omega_t) \tag{2}$$

and

$$\theta_{t+1} = \theta_t + [\nabla_\theta \pi_\theta(s)\nabla_a A^\pi(s, a; \theta_t)|a = \pi(s)]. \tag{3}$$

Thus the update for weights (or dynamic equation) is

$$\frac{d\omega}{dt} = \alpha[r(t) + \gamma V^\pi(p(s_t, a_t); \omega) - V^\pi(s_t; \omega)]\nabla_\omega V^\pi(s_t; \omega)). \tag{4}$$

Under the asynchronous update method, Eq. (4) can be put as

$$\frac{d\omega}{dt} = \alpha\frac{1}{TN}\sum_{i=0}^{N}\sum_{t=0}^{T}[r_{it} + \gamma V^\pi(p(s_{it}, a_{it}); \omega) - V^\pi(s_{it}; \omega)]\nabla_\omega V^\pi(s_{it}; \omega)). \tag{5}$$

Liu *et al.* gives the $Q$-value parameter dynamics under experience buffer [4]. We can rewrite it into $V$-value parameter dynamics under experience buffer

$$\frac{d\omega}{dt} = \beta\int_{t-n_t}^{t}[r(t) + \gamma V^\pi(p(s_t, a_t); \omega) - V^\pi(s_t; \omega)]\nabla_\omega V^\pi(s_t; \omega)). \tag{6}$$

---

[*]Joint first authors.

Taking the discrete approximation, we get

$$\beta \int_{t-n_t}^{t} [r(t) + \gamma V^\pi(p(s_t, a_t); \omega) - V^\pi(s_t; \omega)] \nabla_\omega V^\pi(s_t; \omega)) \tag{7}$$

$$\approx \beta \sum_{i=t-n_t}^{t} [r(i) + \gamma V^\pi(p(s_i, a_i); \omega) - V^\pi(s_i; \omega)] \nabla_\omega V^\pi(s_i; \omega)). \tag{8}$$

The Eq. (5) can be reformed as

$$\alpha \frac{1}{NT} \sum_N \sum_T [r(t) + \gamma V^\pi(p(s_t, a_t); \omega) - V^\pi(s_t; \omega)] \nabla_\omega V^\pi(s_t; \omega)) \tag{9}$$

$$= \alpha \frac{1}{NT} \sum_{i=0}^{NT-1} [r(i) + \gamma V^\pi(p(s_i, a_i); \omega) - V^\pi(s_i; \omega)] \nabla_\omega V^\pi(s_i; \omega)). \tag{10}$$

Therefore Eq. (8) coincides with Eq. (10) when

$$\alpha \frac{1}{NT} = \beta \text{ and } NT - 1 = n_t. \tag{11}$$

Obviously, there always exists $\beta$ and $n_t$ satisfying Eq. (11). This means that our asynchronous policy gradient can approximate the experience replay method.

## B. Solving the $V$-value function directly

In this section we will discuss an alternative way to represent the $V$-value function.

We have already known that the equation of the V-value function is $V^\pi(s) = \underset{s \sim s_0, P_T \sim \pi}{E} [r_T]$ and that of the TD error $\delta = r_t + \gamma V^\pi(p(s_t, a_t)) - V^\pi(s_t)$. We can simplify the TD error as

$$\delta = \begin{cases} D(P_T(s_0, \mathcal{A})) - \gamma V(s_t), & t = T - 1 \\ \gamma V(p(a_t, s_t)) - V(s_t), & t < T - 1. \end{cases} \tag{12}$$

The ideal $V(s_t)$ is to minimize $\frac{1}{2} E[\delta^2]$. De facto, Eq. (12) can be solved directly. Setting $\frac{1}{2} E[\delta^2]$ to be zero, we have

$$\frac{1}{2} E[\delta^2] = 0. \tag{13}$$

Approximately, one of its solution is to make $\delta$ close to zero, thus resulting in

$$\begin{array}{ll} D(P_T(s_0, \mathcal{A})) = \gamma V(s_{t-1}), & t = T \\ \gamma V(p(a_t, s_t)) = V(s_t), & t < T. \end{array} \tag{14}$$

Eq. (14) above can be re-written in unified form

$$V(s_t) = D(P_T(s_0, \mathcal{A})) * \gamma^{-t+T} \tag{15}$$

and

$$Q(s_t, a_t) = r_t + \gamma V^\pi(p(s_t, a_t)) = D(P_T(s_0, \mathcal{A})) * \gamma^{-t+1+T}. \tag{16}$$

Therefore, the advantage function can be written as

$$A(s_t, a_t) = Q(s_t, a_t) - V(s_t) \tag{17}$$

$$= D(P_T(s_0, \mathcal{A})) * \gamma^{-t+1+T} - D(P_T(s_0, \mathcal{A})) * \gamma^{-t+T} \tag{18}$$

$$= D(P_T(s_0, \mathcal{A})) * (\gamma - 1) * \gamma^{-t+T}. \tag{19}$$

Since $P_T(s_0, \mathcal{A})$ contains future information difficult to obtain. If the sub-images are not so many, in practical, we can use the intermediate state $s_t^l$ to approximate the final step $s_t^l \approx P_T(s_0, \mathcal{A})$.

One interesting thing is that if we regard the discriminator $D$ as the reward function, then the $A(s_t, a_t)$ is like the potential-based reward shaping function [6, 1]

$$F = \gamma * r_{t+1}^\gamma - r_t^\gamma \tag{20}$$

$$= \gamma^{T+1} * D(P_T(s_0, \mathcal{A})) - D(P_T(s_0, \mathcal{A})) \tag{21}$$

$$= \gamma^{T+1} * (D(P_T(s_0, \mathcal{A}))) * \gamma^{-t+1} - D(P_T(s_0, \mathcal{A})) * \gamma^{-t}) \tag{22}$$

$$= \gamma^{-t-1+T} * (V(p(s_t, a_t)) - V(s_t)) = \gamma^{-t+1+T} A^\pi(s_t, a_t). \tag{23}$$

Through Eq. (23), we can see that the advantage function $A^\pi$ is the same form as reward shaping function $F$. Moreover, since $A^\pi(s_t, a_t) \propto -D(P_T(s_0, \mathcal{A}))$, this algorithm can be explained as an actor-only reinforcement learning which greedily maximizes the reward shaping function $F$. We use this method to solve sparse reward problem.

## C. Generative loss and policy gradient

From the equations given above, we have two forms of the value function. Now, we will discuss how these two value functions affect the policy gradient method.

### C.1 Solving form of value function directly

If value function is replaced by $D$, the policy gradient can be

$$\underset{s \sim \rho}{E} [\nabla_\theta \pi_\theta(s) \nabla_a A^\pi(s, a) | a = \pi(s)] = \underset{s \sim \rho}{E} [\nabla_\theta \pi_\theta(s) \nabla_a D^\pi(s, a) | a = \pi(s)] * C, \tag{24}$$

where $C = (\gamma - 1) * \gamma^{-t+T}$. This is one form of our value function. We solve it directly and thus can avoid building another neural network to approximate value function. For Generative Adversarial Network (GAN), the original generator loss function is

$$L_G = -E_{x \sim p_g}[D]. \tag{25}$$

The differential form of $L_G$ gives

$$\nabla_{\theta_g} L_G = -\nabla_{\theta_g} E_{x \sim p_g}[D(G_{\theta_g}(x))] = -E_{x \sim p_g}[\nabla_{\theta_g} G_{\theta_g}(x) \nabla_y D(y) | y = G_{\theta_g}(x)]. \tag{26}$$

Eq. (24) and Eq. (26) are of the same form (only different by a decay parameter). Thus using the policy gradient to approximate the original derivative is reasonable.

### C.2 Value function approximated by neural networks

We know that the policy gradient is

$$\underset{s \sim \rho}{E} [\nabla_\theta \pi_\theta(s) \nabla_a A^\pi(s, a) | a = \pi(s)]. \tag{27}$$

According to Eq. (25), the differential form of $L_G$ is

$$\underset{s \sim \rho}{E} [\nabla_\theta \pi_\theta(s) \nabla_a A^\pi(s, a) \nabla_A D^\pi(A) | a = \pi(s)]. \tag{28}$$

Supposing that Eq. (12) approximately equals zero. Then $\nabla_A D(A) = \frac{1}{C_1}$, where $C_1$ is constant which equals to $(\gamma - 1) * \gamma^{-t+T}$. If we regard the $C_1$ as one part of the total learning rate, then we can draw the conclusion that

$$\underset{s \sim \rho}{E} [\nabla_\theta \pi_\theta(s) \nabla_a A^\pi(s, a) | a = \pi(s)] = C_1 * \underset{s \sim \rho}{E} [\nabla_\theta \pi_\theta(s) \nabla_a A^\pi(s, a) \nabla_A D^\pi(A) | a = \pi(s)]. \tag{29}$$

Where $C_1 \underset{s \sim \rho}{E} [\nabla_\theta \pi_\theta(s) \nabla_a A^\pi(s, a) \nabla_A D^\pi(A) | a = \pi(s)] = -C_1 \nabla_{\theta_g} L_G$. Therefore, we can also view this problem from another perspective. If we replace the discriminator $D(G_{\theta_g}(x))$ by advantage function $A^\pi(s, a)$, then Eq. (27) is the same as Eq. (26). This means that for the neural-network-based value function, it acts as the discriminator. Moreover, the advantage function $A^\pi(s, a)$ is trained through the discriminator. From this perspective, we think that the advantage function and the discriminator bear a resemblance. Thus, the equation $\nabla_\theta J(\pi_\theta) \approx C * \nabla_\theta L_{G_\theta}$ is established, where $C = -C_1$ is a positive constant.

**D. Details on exposure blending**

In this section, we will discuss how to get the fusion weight matrix and its meaning.

**D.1 Fusion weight matrix**

For different locally-exposed images $\{S_1^g, \ldots, S_n^g\}$, there always exists some areas that are over-exposed, under-exposed, or both. If we just combine them according to the segmentation directly, there would be seams at the boundary of each segment. In order to solve this problem, we introduce an exposure blending method.

Firstly, we should define which parts of an image are well-exposed. Similar to Mertens *et al.* [5], we fulfill this through three evaluation metrics: contrast, saturation, and well-exposedness score.

We use Laplacian filter to the grayscale version of an image to calculate the contrasts; and leverage the standard deviations within R, G and B channels of each pixel to calculate the saturation; and the distance of raw intensities to 0.5 is to calculate the well-exposedness score. The final weight function can be

$$W^k = (L(Gray(S_i^k)))^{C_1} * (std(S_i^k))^{C_2} * (Gauss\_cur(S_i^k))^{C_3}, \tag{30}$$

where $Gray(\cdot)$ is to turn the image into grayscale, $L(\cdot)$ is the Laplacian pyramid decomposition function, $std(\cdot)$ is to get standard deviation within the R, G and B channels at each pixel, $Gauss\_cur(\cdot)$ means Gaussian function with mean $= 0.5$ and standard deviation $= 0.2$, and $\{C_1, C_2, C_3\}$ are the hyper-parameters to define which evaluation is significant.

After we get the weight matrix, it is easy to calculate the blending results by

$$L(S_o^{ij})^l = \sum_{k=1}^{n} Gauss(w_{ij}^l)^k L(S_{ijl}^g)^k. \tag{31}$$

In fact, the adjustment of our algorithm based on local exposures can be seen as creating a sequence of pseudo over-exposed and under-exposed images and then fusing them together. Li *et al.* propose a method of generating differently virtual exposed images to perform the single image brightening [3]. From this perspective, our method can also be regarded as an adaptive single photo fusion method which can choose exposure values and exposure areas in images.

**E. Algorithm pseudocodes**

In this section, we will present the detailed procedures of our algorithms. Algorithm 1 is the DeepExposure algorithm with neural-network-approximated $V$ function. Algorithm 2 is the one to leverage the discriminator as $V$ function and Algorithm 3 is how to use our method to high-resolution images of any size (the inference stage).

**F. Network structures and test process**

In this section, we demonstrate the details of network structures and the test process in Figure 1 and Figure 2.

**G. Experiment supplements**

In this section, we will give more examples of our results from different sizes and different scenarios. Moreover, the detailed process is also exhibited in this section.

**G1. Large image results**

In this section we will demonstrate our results on large images.

From the comparison of Figure 3 and Figure 4, we can find that our method preserves the facial details of original images (Figure 5). Figure 6 and Figure 7 show that our method can preserve the details of texts in the image better, which is useful in some areas, like taking notes by a photo where we need to read the words.

These experiments prove that our method has the advantage to deal with large-size images with well-preserved details of original images.

**Algorithm 1 DeepExposure I**: image retouching with neural-network-approximated $V$ function

---

Initialize the policy network $\pi^\theta$, the value network $V^\omega$, and the discriminator network $D^\beta$ with random weights $\theta$, $\omega$ and $\beta$, respectively.

Set the pre-training stage: p-step = 30, the policy-training stage step: g-step = 1 and the discriminator-training stage step: d-step = 5.

Build a memory buffer to restore retouched photos $M_1$, an expert dataset to restore the expert-retouched photos $M_2$ and a raw photo dataset $M_3$.

Initialize an Ornstein-Uhlenbeck process $OU$.

**for** p-step **do**

    choose some raw photos from $M_3$;

    create retouched photos by roll-out method on raw photos;

    choose arbitrary expert-retouched photos from the expert dataset $M_2$;

    pre-train the discriminator network $D^\beta$ using Eq. (12) **in the main text** with unpaired machine-retouched photos and expert-retouched photos;

**end for**

**repeat**

    **for** $j$ in $g\_step$ **do**

        choose a minibatch of raw images from $M_3$;

        **for** $k$ in $minibatch\_size$ **do**

            segment the image into sub-images;

            **for** $t$ in $\{1, \ldots, T\}$ **do**

                **if** is the first sub-image $seg_0$ **then**

                    $e_0 = \pi^\theta(S_0, seg_0, S_0) + OU$   // $e_0$ is the exposure and $S_0$ is the original input image

                    $S_1^g = global\_filter(e_0, S_0), \; S_1^l = local\_filter(e_0, S_0)$

                **else**

                    $e_{t+1} = \pi^\theta(S_0, seg_t, S_t^l) + OU$

                    $S_{t+1}^g = global\_filter(e_t, S_0), S_{t+1}^l = local\_filter(e_t, S_t)$

                **end if**

                calculate the one batch gradients of $\theta$ and $\omega$, and collect them;

                collect global $S_{t+1}^g$;

            **end for**

        **end for**

        perform exposure fusion using $\{S_1^g, \ldots, S_T^g\}$ Eq. (30) in the supplementary materials and Eq. (16) **in the main text** ;

    **end for**

    update the policy network $\pi^\theta$ and the value network $V^\omega$ using Eq. (11) and Eq. (10) **in the main text** ;

    **for** $i$ in $d\_step$ **do**

        choose arbitrary expert-retouched photos from the expert dataset $M_3$;

        choose a batch of retouched photos from memory $M_1$;

        update the discriminator network $D^\beta$ using Eq. (12) **in the main text** ;

        delete those data in memory $M_1$;

    **end for**

**until** all networks converge

---

---

**Algorithm 2 DeepExposure II**: image retouching with the discriminator as the value function

---

Initialize the policy network $\pi^\theta$ and the discriminator network $D^\beta$ with random weights $\theta$ and $\beta$.
Set the pre-training stage: p-step = 30, the policy-training stage step: g-step = 1 and the discriminator-training stage step: d-step = 5.
Build a memory buffer to restore retouched photos $M_1$, an expert dataset to restore the expert-retouched photos $M_2$ and a raw photos dataset $M_3$.
Initialize an Ornstein-Uhlenbeck process $OU$.
**for** p-step **do**
    choose some raw photos from $M_3$;
    create retouched photos by roll-out method on raw photos;
    choose arbitrary expert-retouched photos from the expert dataset $M_2$;
    pre-train the discriminator network $D^\beta$ using Eq. (12) **in the main text** with unpaired machine-retouched photos and expert-retouched photos;
**end for**
**repeat**
    **for** $j$ in $g\_step$ **do**
        choose a minibatch of raw images from $M_3$;
        **for** $k$ in $minibatch\_size$ **do**
            segment the image into sub-images;
            **for** $t$ in $\{1,\dots,T\}$ **do**
                **if** is the first sub-image $seg_0$ **then**
                    $e_0 = \pi^\theta(S_0, seg_0, S_0) + OU$    // $e_0$ is the exposure and $S_0$ is the original input image
                    $S_1^g = global\_filter(e_0, S_0),\ S_1^l = local\_filter(e_0, S_0)$
                **else**
                    $e_{t+1} = \pi^\theta(S_0, seg_t, S_t^l) + OU$
                    $S_{t+1}^g = global\_filter(e_t, S_0), s_{t+1}^l = local\_filter(e_t, S_t)$
                **end if**
                calculate the one-batch gradient of $\theta$ and $\omega$, and collect them;
                collect global $S_{t+1}^g$;
            **end for**
        **end for**
        perform exposure fusion using $\{S_1^g,\dots,S_T^g\}$ according to Eq. (30) in the supplementary materials and Eq. (16) **in the main text** ;
    **end for**
    calculate the advantage function using Eq. (19);
    update the policy network $\pi^\theta$ using Eq. (10) **in the main text**;
    put the retouched photos into memory $M_1$;
    **for** $i$ in $d\_step$ **do**
        choose arbitrary expert-retouched photos from the expert dataset $M_3$;
        choose a batch of retouched photos from memory $M_1$;
        update the discriminator network $D^\beta$ using Eq. (12) **in the main text**;
        delete those data in memory $M_1$;
    **end for**
**until** all networks converge

---

**Algorithm 3 DeepExposure**: the test stage
___
**Require:** A raw photo $I_0$ with any size and a well-trained policy network.
  Create a small version of that photo by resize $I_0$ into $64 \times 64$ as $S_0$.
  Segment $S_0$ into sub-images.
  **for** $t$ in $\{1, \ldots, T\}$ **do**
    **if** is the first sub-image $seg_0$ **then**
      $e_0 = \pi^\theta(S_0, seg_0, S_0)$   // $e_0$ is the exposure and $S_0$ is the original low-resolution input image

      $S_1^g = global\_filter(e_0, S_0), \; S_1^l = local\_filter(e_0, S_0)$
      $I_1^g = global\_filter(e_0, I_0)$
    **else**
      $e_{t+1} = \pi^\theta(S_0, seg_t, S_t^l)$
      $S_{t+1}^g = global\_filter(e_t, S_t), \; S_{t+1}^l = local\_filter(e_t, S_t)$
      $I_{t+1}^g = global\_filter(e_t, I_t)$
    **end if**
    collect global $I_{t+1}^g$;
  **end for**
  perform exposure fusion using $\{I_1^g, \ldots, I_T^g\}$ according to Eq. (30) and Eq. (31) in the paper;
  **return** retouched photo
___

## G2. The detailed process

We will show some retouching process here.

Figure 9 and Figure 10 reveal the main principle how the machine learns: For different segmentations with different colors and brightness, the main purpose is to expose the original image to a certain level that can facilitate the effect of fusion.

## G3. More compared results

Here, we exhibit more results of our experiments from Figure 11 to Figure 50. The compared results contain original images, our DeepExposure I, expert C as well as the state-of-the-art sequence-based method of Exposure [2].

Figure 1: The testing process. The testing process is divided into three stages: image pre-processing, acquisition of operation parameters as well as raw low-resolution image operations. The image pre-processing stage consists of image segmentation. In the stages of acquiring operation parameters, each exposure value $e_t$ is obtained by the input state $s_t = \{seg_t, S_0, S_t^l\}$: the $t$-th sub-image, the original image, and the direct fusion result. At the first stage, since there is no fusion, we use the original image to replace $S_t^l$. The local filter is employed to perform the exposure on local area while the global fusion operates on global area. The following steps are the same. In the third step, after collecting all the exposure operations, the global filter will operate on the original raw large image $I_0$ and the resulting re-exposed images are fused together to get the final result.

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

Figure 3: The retouched image of our method. The details of the texts are shown in the bottom page.

Figure 4: The retouched image of DPED method. The details of the texts are shown in the bottom page.

Figure 5: The original large size image.

Figure 6: The retouched image of our method. The facial details are shown in the bottom page.

Figure 7: The retouched image of DPED method. The facial details are shown in the bottom page.

Figure 8: The original large size image.

Figure 9: The process of our retouching stage.

Figure 10: The process of our retouching stage.

| RAW Input | Expert C | Ours | Exposure |
|---|---|---|---|

Figure 11: The compared results.

RAW Input      Expert C      Ours      Exposure

Figure 12: The compared results.

RAW Input      Expert C      Ours      Exposure

Figure 13: The compared results.

| RAW Input | Expert C | Ours | Exposure |
|:---:|:---:|:---:|:---:|

Figure 14: The compared results.

| RAW Input | Expert C | Ours | Exposure |

Figure 15: The compared results.

| RAW Input | Expert C | Ours | Exposure |
|-----------|----------|------|----------|

Figure 16: The compared results.

| RAW Input | Expert C | Ours | Exposure |

Figure 17: The compared results.

| RAW Input | Expert C | Ours | Exposure |
|---|---|---|---|

Figure 18: The compared results.

RAW Input      Expert C      Ours      Exposure

Figure 19: The compared results.

RAW Input            Expert C            Ours            Exposure

Figure 20: The compared results.

| RAW Input | Expert C | Ours | Exposure |
|-----------|----------|------|----------|

Figure 21: The compared results.

| RAW Input | Expert C | Ours | Exposure |
|---|---|---|---|

Figure 22: The compared results.

RAW Input  Expert C  Ours  Exposure

Figure 23: The compared results.

RAW Input          Expert C          Ours          Exposure

Figure 24: The compared results.

RAW Input          Expert C            Ours              Exposure

Figure 25: The compared results.

| RAW Input | Expert C | Ours | Exposure |
|-----------|----------|------|----------|

Figure 26: The compared results.

RAW Input  Expert C  Ours  Exposure

Figure 27: The compared results.

| RAW Input | Expert C | Ours | Exposure |
|:---:|:---:|:---:|:---:|

Figure 28: The compared results.

RAW Input　　　Expert C　　　Ours　　　Exposure

Figure 29: The compared results.

RAW Input&emsp;&emsp;&emsp;&emsp;Expert C&emsp;&emsp;&emsp;&emsp;Ours&emsp;&emsp;&emsp;&emsp;Exposure

Figure 30: The compared results.

RAW Input          Expert C          Ours          Exposure

Figure 31: The compared results.

RAW Input　　　Expert C　　　Ours　　　Exposure

Figure 32: The compared results.

Figure 33: The compared results.

| RAW Input | Expert C | Ours | Exposure |
|---|---|---|---|

Figure 34: The compared results.

RAW Input            Expert C            Ours            Exposure

Figure 35: The compared results.

RAW Input     Expert C     Ours     Exposure

Figure 36: The compared results.

RAW Input      Expert C      Ours      Exposure

Figure 37: The compared results.

RAW Input          Expert C          Ours          Exposure

Figure 38: The compared results.

RAW Input          Expert C            Ours            Exposure

Figure 39: The compared results.

RAW Input          Expert C          Ours          Exposure

Figure 40: The compared results.

Figure 41: The compared results.

RAW Input      Expert C      Ours      Exposure

Figure 42: The compared results.

| RAW Input | Expert C | Ours | Exposure |
|:---:|:---:|:---:|:---:|

Figure 43: The compared results.

RAW Input　　　　Expert C　　　　　Ours　　　　Exposure

Figure 44: The compared results.

RAW Input       Expert C       Ours       Exposure

Figure 45: The compared results.

RAW Input       Expert C       Ours       Exposure

Figure 46: The compared results.

RAW Input       Expert C       Ours       Exposure

Figure 47: The compared results.

RAW Input    Expert C    Ours    Exposure

Figure 48: The compared results.

| RAW Input | Expert C | Ours | Exposure |
|-----------|----------|------|----------|

Figure 49: The compared results.

RAW Input    Expert C    Ours    Exposure

Figure 50: The compared results.