[Reviews · NeurIPS 2018]

Reviewer 1



The paper presents a novel reinforcement adversarial learning approach for image enhancement. The method parameterizes image enhancement operations in local regions that are analyzed separately and then merged together to produce the final version. The method learns to apply local enhancements as part of a decision process optimized with reinforcement learning. In order to provide feedback to the enhancement agent, the method learns a reward function in an adversarial setting: images enhanced by the agent are discriminated from images enhanced by expert photographers. Since the image generator is a non-differentiable function, the policy network and the discriminator are trained separately in an alternating fashion using asynchronous updates and unpaired data samples, respectively. The overall strategy is very interesting, the formulation of the solution is sound, and the results show consistent improvements over baseline methods. Comments: * The organization of the paper is good, and the flow is generally clear. However, writing style and grammar needs to be improved in general throughout the manuscript. * In Table 1, it is not clear which method is Ours 1 and Ours 2. The main text does not seem to clarify either. Please explain. * Equations 7 to 9 seem to be the standard equations for DPG. Are these details necessary in the discussion? Maybe the space can be used to discuss more results or information specific to this paper. * Line 93: s_{t+1} = a_t o s_t = p(a_t, s_t). The function $p$ has the parameters in the opposite order wrt equations 5 and 6.

Reviewer 2



This paper proposes a novel method for optimal exposure operation in low quality images. The method uses reinforcement learning coupled with a discriminant loss (from GANs) to learn the optimal sequence of operations (i.e., the different exposures for each subimage component from a semantic segmentation of the input image) that generate, through a blender of all the components, a good quality - better exposed image. The main concern with this paper is the poor clarity of exposition. In particular: (i) The motivation and formulation of the problem. The formal definition of the image processing problem is lacking. Section 2.1. introduces the reinforcement learning approach to the problem but in a chaotic way. Semantic segmentation is one major component but it's not discussed. (ii) Limited novelty. No new reinforcement learning method is introduced. The paper is is an application of already known tools, to an image processing problem. Since this problem has been addressed before using also RL tools [13] the novelty is very limited. (iii) Relation to previous work is not clear. Notably to [13]. Is this just a slight improvement? What are the differences? (iv) The description, justification and analysis of the proposed algorithm is not clearly presented and discussed (Section 2) v) Evaluation of the algorithm needs to be better. Sliced analysis are needed. Average performance in a whole dataset may hide important aspects (Table 1, very small diferences in PSNR...). Breaking down performance and a deeper analysis of the results is needed. In short, the experimental section failed to convince me that this approach is doing better than the rest. (vi) The use of English needs a severe cleanup. In my opinion, this work is still very premature. Its contributions are not clear (in particular its novelty); a better organization with an improved narrative, and a deeper analysis need to be incorporated before one can properly judge the real contributions. **AFTER REBUTTAL** I thank the authors for their response. The authors did a good job clarifying many of the raised points in the reviews. Unfortunately, I still have major problems with this paper. This paper is about an image retouching method, as such, its quantitative/qualitative evaluation is poor. It is not clear how good is its performance compared to previous work. Moreover, the major contribution is that the method produces local adjustments (instead of global [13]). For this, the method relies on an image segmentation algorithm (that is plugged and is not part of the proposed method). How robust is the method to a wrong segmentation? Retouching locally might produce "semantic" artifacts (e.g., a part of a body is modified differently than the rest of the body). None of this is discussed. So the major contribution of the paper is an application of reinforcement learning to an image processing problem. There are a few technical aspects (non-differentiable, adversarial learning) that are interesting but not really new. I thus think this is preliminary work and the authors need to do more analysis to convince the reader that this method produces good quality (better than the others) results.

Reviewer 3



The paper proposes an image retouching method in a local processing and fusion manner. The method first segments the image into sub-images, then each sub-image is passed into a reinforment-learning-based method to output an optimal exposure for this region. The exposure-corrected images under all the exposures (estimated from all the sub-images) are fused through a blending approach. Pros: 1. The authors propose an image retouching method by exploiting local adjustment instead of global adjustment. This could better handle the cases that bright and dark regions appear in the same picture. The three components are reasonably assembled. 2. The proposed method achieves good quantitative result in terms of PSNR. Cons: 1. The proposed method consists of three components, segmentation, local retouching, and fusion. The authors use the method [8] for segmentation, and the method [24] for blending. And the retouching strategy is very similar to [13], which also uses reinforcement-learning to update the exposure and a discriminator to represent the loss/AE function (and approximating the loss using CNN). The major difference is that [13] applies global adjustment instead of local one. Considering those, the proposed method is more a systematic work by combining several existing components, thus marginalize its novelty. If there are other differences between [13] and the local retouching component of this method (which I did not see much), the authors should provide a comparison by replacing the component with the method in [13], to demonstrate the advantages of the proposed one. 2. The authors provide the quantitative results on MIT-Adobe 5K in terms of PSNR. As there are no ground-truth images for this task, are the expert retouched results treated as the reference? If so, the results using this metric is less informative, as the quantitative evaluation with respect to the expect retouched results could be biased. Since the task is subjective, a user study to evaluate the results is suggested. 3. From the quantitative results, the improvement over [13] is little and less than expected (less than 0.06db considering [13] also use the approximation function). I would like to see more analysis on this. 4. In Figure 3, the results using [5] look better than the proposed method. Some more analysis on this and quantitative comparison with [5] are recommended. 5. In Figure 3, the results using the proposed method look over-exposed. What is the reason for that? Is it because the exposed images are favored by the discriminator? I would like to see some analysis on that. In general, the novelty of the paper is limited and the experimental validation is a bit lacking. ---------- The rebuttal addresses some of my mis-understanding before, especially on the non-differentiable form and the generative loss approximation. And therefore the usage of asynchronous reinforcement learning makes sense. In this sense, the proposed method is technically sound and novel in terms of the modification on the local adjustment. For the comparison to [13] with local adjustment in the rebuttal, I don't quite understand the setup (the setup still feels like the proposed framework). And the provided quantitative is hard to explain, which is much worse (0.36db lower) than global retouching itself. A more reasonable way is directly applying the global retouching [13] on different segments to estimate their exposure settings (differentiable loss directly on the segment) and then merge all the retouched results of different exposure settings via the exposure fusion. Since there are a few modifications and there lacks of ablation study (and difficult to evaluate) for each small modification. I think this experiment at least helps understand whether the improvement is from simply local adjustment or others.